# S100B Protein as a Therapeutic Target in Multiple Sclerosis: The S100B Inhibitor Arundic Acid Protects from Chronic Experimental Autoimmune Encephalomyelitis

**DOI:** 10.3390/ijms222413558

**Published:** 2021-12-17

**Authors:** Chiara Camponeschi, Maria De Carluccio, Susanna Amadio, Maria Elisabetta Clementi, Beatrice Sampaolese, Cinzia Volonté, Maria Tredicine, Vincenzo Romano Spica, Rosa Di Liddo, Francesco Ria, Fabrizio Michetti, Gabriele Di Sante

**Affiliations:** 1Section of General Pathology, Department of Translational Medicine and Surgery, Università Cattolica del Sacro Cuore, Largo Francesco Vito 1, 00168 Rome, Italy; chiaracamponeschi94@gmail.com (C.C.); maria.decarluccio01@gmail.com (M.D.C.); maria.tredicine@unicatt.it (M.T.); gabriele.disante@unicatt.it (G.D.S.); 2Department of Neuroscience, Università Cattolica del Sacro Cuore, Largo Francesco Vito 1, 00168 Rome, Italy; 3IRCCS Santa Lucia Foundation, Via del Fosso di Fiorano 65, 00143 Rome, Italy; s.amadio@hsantalucia.it (S.A.); c.volonte@hsantalucia.it (C.V.); 4Istituto di Scienze e Tecnologie Chimiche “Giulio Natta” SCITEC-CNR, Largo Francesco Vito 1, 00168 Rome, Italy; elisabetta.clementi@scitec.cnr.it (M.E.C.); beatrice.sampaolese@scitec.cnr.it (B.S.); 5National Research Council, Institute for Systems Analysis and Computer Science, Via dei Taurini 19, 00185 Rome, Italy; 6Department of Movement, Human and Health Sciences, Laboratory of Epidemiology and Biotechnologies, University of Rome “Foro Italico”, Piazza Lauro De Bosis 6, 00135 Rome, Italy; vincenzo.romanospica@uniroma4.it; 7Department of Pharmaceutical and Pharmacological Sciences, University of Padua, Via Marzolo 5, 35131 Padua, Italy; rosa.diliddo@unipd.it; 8Department Laboratory and Infectious Diseases Sciences, Fondazione Policlinico Universitario, A. Gemelli IRCCS, Largo Agostino Gemelli 1–8, 00168 Rome, Italy; 9IRCCS San Raffaele Scientific Institute, Università Vita-Salute San Raffaele, via Olgettin 60, 20121 Milan, Italy; 10Department of Surgery and Medicine, Institute of Human, Clinical and Forensic Anatomy, Piazza L. Severi 1, 06125 Perugia, Italy

**Keywords:** S100B inhibitor, multiple sclerosis, experimental autoimmune encephalomyelitis, arundic acid

## Abstract

S100B is an astrocytic protein behaving at high concentration as a damage-associated molecular pattern molecule. A direct correlation between the increased amount of S100B and inflammatory processes has been demonstrated, and in particular, the inhibitor of S100B activity pentamidine has been shown to ameliorate clinical scores and neuropathologic-biomolecular parameters in the relapsing-remitting experimental autoimmune encephalomyelitis mouse model of multiple sclerosis. This study investigates the effect of arundic acid (AA), a known inhibitor of astrocytic S100B synthesis, in the chronic experimental autoimmune encephalomyelitis, which is another mouse model of multiple sclerosis usually studied. By the daily evaluation of clinical scores and neuropathologic-molecular analysis performed in the spinal cord, we observed that the AA-treated group showed lower severity compared to the vehicle-treated mice, particularly in the early phase of disease onset. We also observed a significant reduction of astrocytosis, demyelination, immune infiltrates, proinflammatory cytokines expression and enzymatic oxidative reactivity in the AA-treated group. Overall, our results reinforce the involvement of S100B in the development of animal models of multiple sclerosis and propose AA targeting the S100B protein as a focused potential drug to be considered for multiple sclerosis treatment.

## 1. Introduction

Multiple sclerosis (MS) is an autoimmune disease characterized by demyelination and axonal loss. The pathologic hallmarks of the disease are focal lesions/plaques which are scattered throughout the central nervous system (CNS); these plaques contain a combination of pathological features, including edema, inflammation, gliosis, demyelination, and/or axonal loss. A variety of pathogenic processes have been implicated in plaque formation, including oxidative stress promoted by macrophages/microglia, neurotoxic factors secreted by activated T cells, and autoantibodies directed at self-antigens. However, the precise mechanisms leading to inflammation are largely unknown; hence, the identification of such factors would be essential to develop target therapeutic strategies for the disease [1,2].

S100B is a small EF-related Ca^2+^ and Zn^2+^-binding protein (a helix−loop−helix structure is formed by the E and F helices), which is mainly synthesized by astrocytes in the nervous system and, to a lesser extent, by oligodendrocytes. It exerts both intracellular and extracellular actions. Although a clear and univocal intracellular function for this protein has not been delineated, astrocyte-secreted S100B is regarded to act as an autocrine, paracrine, or even endocrine factor with concentration-dependent effects. In physiological conditions, astrocytes secrete S100B, which exerts a neurotrophic action at nanomolar concentration; under stress conditions, including nervous tissue inflammation, astrocytes secrete S100B at micromolar concentration, thus exerting a neurotoxic effect and behaving as a danger/damage-associated molecular pattern (DAMP) molecule. S100B mainly interacts with surrounding cell types through the activation of the receptor for advanced glycation endproducts (RAGE), a ubiquitous, transmembrane immunoglobulin-like receptor that acts as both an inflammatory intermediary and a critical inducer of oxidative stress. Data have been reported, indicating S100B as a key molecule in pathogenic processes and S100B levels in biological fluids as a reliable biomarker in a series of neural disorders, including acute brain injury, Alzheimer’s disease, Parkinson’s disease, or amyotrophic lateral sclerosis. Interestingly, in many cases, data have shown that the overexpression/administration of the protein induces the worsening of the disease, while its deletion/inactivation produces the amelioration of these disorders [3,4,5].

Elevated levels of S100B are detected in cerebrospinal fluid (CSF) [6] and sera [7] of MS patients in the acute phase, being reduced in the stationary phase of the disease. In addition, an increased expression of S100B is detected in both active demyelinating and chronic active MS plaques [8]. In ex vivo demyelinating models, a marked astrocytic elevation of S100B is observed upon demyelination, while the inhibition of S100B action reduces demyelination and downregulates the expression of inflammatory molecules [7]. The blockade of RAGE suppresses demyelination in a rodent demyelinating model of experimental autoimmune encephalomyelitis (EAE) [9], and the S100B/RAGE axis plays a crucial role in oligodendrocyte myelination processes [10]. More recently, we have demonstrated that pentamidine (PTM), an approved antiprotozoal drug known to block S100B activity [11,12], ameliorates clinical disease scores and neuropathologic and biomolecular parameters in the relapsing-remitting experimental autoimmune encephalomyelitis SJL/J mouse model of MS [13].

This suggests that S100B neutralization emerges as a promising therapeutic target in MS. In order to further verify this hypothesis, we used arundic acid (AA; ONO 2506), a known inhibitor of astrocytic S100B synthesis [14], in an chronic progressive MS model (P-EAE mice) [15], which reproduces a severe form of MS with slow continuous worsening of neurological decline. At present, people affected by this form of MS can only benefit from symptomatic therapy [16,17].

## 2. Results

### 2.1. Clinical Results

Figure 1 shows the effect of the treatment with AA on the clinical evolution of EAE in C57Bl/6 mice (chronic MS model). As indicated by the timeline (Appendix A), AA (4 mg/kg, intraperitoneal) was administered from day 7 after immunization [18,19]. Significant differences in clinical score (Mann−Whitney U test, *t*-test, and *p*-values are indicated in the figure legend) were observed: the AA-treated group showed lower severity, when the mean and cumulative disease scores were compared to those of the vehicle-treated mice, particularly in the early phase of disease onset. Indeed, after day 9 post-immunization (d.p.i.), the group of vehicle-treated mice (blue line and symbols) begun to show symptoms that significantly persisted from day 10 to day 23. The AA-treated mice (fuchsia line and symbols) delayed disease onset (certain mice never reached a disease score of 1) and led to a significantly milder disease. These data supported the hypothesis that the inhibition of S100B synthesis by AA exerts its efficacy particularly in the acute phase of MS, perhaps behaving as a disease-delaying factor (Figure 1A). These results were reinforced by comparing weight loss after 9 d.p.i. in the vehicle-treated mice and in the AA-treated mice (small box in Figure 1A). We also compared the average score of the disease of each mouse (calculated as the sum of the score of each mouse compared to the number of days of disease; Figure 1B) and the cumulative disease of each individual (which was equivalent to the sum of the score of each mouse for each day of the disease; Figure 1C); as shown, AA treatment reduced both the average score of disease (*p*-value < 0.0001, Mann−Whitney U test, *t*-test; Figure 1B) and the cumulative disease score (*p*-value < 0.0001, Mann−Whitney U test, *t*-test; Figure 1C).

### 2.2. Modulation by AA of Astrocytosis and Demyelination during EAE

We performed a semi-quantification analysis of specific markers of gliosis (S100B^+^ and GFAP^+^) and demyelination (sum of the MBP-negative areas) on C1–C4 cervical spinal cord segments, where neuropathological biomarkers are known to be especially apparent in this MS animal model (for review, see [20]), comparing EAE mice treated with the vehicle or AA.

Treatment with AA consistently modulated astrocytosis, by reducing the number of S100B^+^, as expected, and GFAP^+^ cells (cells/mm^2^; *p*-values of 0.082 and 0.0043 for the vehicle and AA, respectively, Mann−Whitney U test, *t*-test; Figure 2A,B), and decreasing demyelination by preserving MBP^+^ areas (µm^2^; vehicle vs. AA, *p* = 0.051 Mann–Whitney, *t*-test; Figure 2C). Figure 2A–C shows the most representative 5× and 20× immunohistochemical (IHC) images analyzed for each CNS region of each mouse. We focused specifically on the cervical region of the spinal cord, because pathology in this mouse model is known to be particularly evident in this area [20]. Appendix A shows all the IHC images of this CNS region. According to both IHC staining and immunofluorescent (IF) analysis, we observed clear reductions of S100B, MBP, and GFAP, markers particularly evident in the spinal cord of the AA-treated group. We examined also another area of the CNS (cerebellum), where these differences, although partially detectable, were less important, as expected (Appendix A).

### 2.3. Impact of AA on the CNS Histologic Immune Infiltrates and on Neuroinflammation during EAE

To evaluate the ability of AA to prevent, delay or reduce the immune infiltration (intended as lymphocytic, CD4^+^ cells) and reactivity of microglia and/or circulating monocytes recruited to the CNS (CD68^+^ cells), we examined these two markers in the same CNS areas described above.

The semi-quantitative analysis on the spinal cord (C1–C4) segments revealed that AA treatment was able to significantly decrease CD4^+^ immune infiltrates (cells/mm^2^; *p*-value = 0.0303, Mann−Whitney U test, *t*-test), as reported in Figure 2E. Moreover, as reported in Figure 2F, CD68^+^ reactive microglia significantly decreased after AA administration (cells/mm^2^; *p*-value = 0.0043, Mann−Whitney U test, *t*-test), reinforcing the hypothesis that this treatment had immunomodulatory and anti-(neuro)inflammatory effects.

### 2.4. Regulation of Proinflammatory Cytokines and Enzymatic Oxidative Reactivity by AA

The effects of AA on neuroinflammation were assessed by testing the different expression levels of IL1β and INFγ by real-time PCR (RT-qPCR) on total mRNA. Figure 3 shows the results obtained from the posterior regions of the CNS (namely cerebellum, pons, and bulb) on day 24 after immunization, which are known to be significantly affected in this mouse model [22] and where S100B protein is known to be highly expressed in C57Bl/6 mice [23]. Figure 3A shows a significant reduction of the IL1β gene expression level after treatment with AA compared to that in the vehicle-treated, EAE-affected group, thus confirming our previous reports indicating a role of S100B inhibitors in the modulation of innate immunity-driven neuroinflammation [13]. Of note, AA was not able to modulate INFγ gene expression levels (Figure 3B) as we instead observed after treatment with PTM on the SJL/J strain [13], thus suggesting a potential different mechanism of action of AA and PTM on neuroinflammation. The impact of AA on the expression of S100B gene were also evaluated, and surprisingly, we found no significant variation of S100B mRNA among groups in the posterior region of CNS (Figure 3C; *p*-value = 0.6623, Mann−Whitney U teste, *t*-test). Nevertheless, a decrease of S100B protein levels between the AA-treated and sham-treated mice could be observed, which became significant in non-EAE mice (Figure 3D; *p*-value = 0.329, Mann−Whitney U test, *t*-test). We tested IL1β and S100B gene expression levels from mRNA derived from the spinal cord (C1–C4 segments) of the vehicle- and AA-treated EAE-affected mice, finding results comparable to cerebellum molecular data and IHC spinal cord outcomes (Appendix A).

The determination of reactive oxygen species (ROS) and nitric oxide synthase (NOS) in the posterior CNS of the treated EAE-affected mice compared to in the vehicle group revealed that AA was able to inhibit both activities, although within different extents and statistical significance (ROS: *p* = 0.0043, Mann−Whitney U teste, *t*-test, Figure 3E; NOS: *p* = 0.0519, Mann−Whitney U test, *t*-test, Figure 3F).

## 3. Discussion

Data are shown, indicating that AA ameliorated clinical conditions as well as neuropathological and biomolecular parameters in the animal model of progressive MS in EAE C57Bl/6 mice. The more diffused relapsing-remitting (RR) course of MS is often followed by a phase of insidious worsening of neurologic function independent from relapses that corresponds to progressive MS. Until recently, no treatments with demonstrated efficacy in terms of preventing disability worsening were available for this insidious form of MS [24]. Thus, the individuation of a putative remedy, although at present restricted for its efficacy to animal models, may offer promising perspectives for the therapy of this disease.

Using experimental animal models, AA has also been shown to improve other neural disorders, such as Alzheimer’s disease [25], Parkinson’s disease [26], acute brain, and spinal cord injury [14,19,27,28,29,30,31,32,33]. In these cases, its effect has been essentially attributed to the inhibition of astrocytic S100B synthesis [14,27]. Reasonably, also in the present case, the amelioration of EAE disease course, resulting in a delayed and mild disease course (Figure 1), might be attributed to the effect on astrocytic S100B. This possibility is also suggested by our immunohistochemical results indicating the poor expression of S100B in astrocytes of the AA-treated animals (Figure 2A), in accordance with previous studies indicating that AA reduces the number of cells expressing S100B [18,30]. Indeed, the biomolecular evaluation of S100B in brain tissue showed intriguingly that the protein was not reduced as compared to in the control animals. Similar results were obtained after the administration of AA in neonatal rats where experimental hypoxia ischemia was induced [33]. This result was obtained in whole tissue extracts, where the evaluation of S100B expression in individual cells could not be assessed. In addition, the possibility has also been advanced in this respect that AA acts on S100B secretion during inflammatory phenomena, thus reducing its effects more than on the expression of the protein [34]. While an effect of AA on S100B may reasonably be considered, how this effect is performed if it is accompanied by other phenomena appears still to deserve consideration. It may be relevant in this respect that the overexpression of S100B through its RAGE receptor is regarded to act via the activation of nuclear factor (NF)-kB proinflammatory cascade [35,36]. Reasonably, the antinflammatory effects of AA administration observed in our in vivo experiments were mediated by the interaction with S100B, as already proposed for other in vivo experimental models of neural injury [28]. In the present in vivo experimental model of MS, AA appeared also to modulate immune infiltration (Figure 3), which was not regarded as a peculiarity of this drug. However, while the direct effects of AA on immune infiltrates have not been described, it may be reasonable to attribute this phenomenon to its interference with S100B which, behaving as a DAMP, is regarded to play an immunoregulatory role (for review, see [37]). It may be relevant in this respect that the blockade of the S100B receptor RAGE has been reported to markedly decrease the infiltration of immune cells in the CNS of EAE animals [9]. In more general terms, the action exerted by AA on astrocyte function may reasonably mediate its influence on immune cells [38,39]. In this respect, the effect of AA on astrocytes, involving S100B downregulation, might be regarded as a mechanism leading to immunomodulation. Interestingly, we observed a similar immunomodulatory effect in RR−EAE mice after treatment with the other S100B inhibitor PTM [13], which also ameliorates clinical scores and pathological parameters in this in vivo MS model. Thus, these results pave a way to promising research addressing possible pathways connecting AA, PTM, S100B, and immunomodulation. In any case, clinical trials for the use of the AA to counteract other neural disorders, such as acute brain injury, amyotrophic lateral sclerosis, Alzheimer’s disease, and Parkinson’s disease, have already been performed [40], and interestingly, S100B is regarded to play a crucial role in pathogenic processes of all previously cited disorders [5]. This study reinforces the involvement of S100B in MS processes after our first demonstration in vivo that interference with S100B activity ameliorates RR-EAE [13] and is a prerequisite step towards clinical trials addressing the use of AA in MS.

The block of S100B action using PTM, which is regarded to inhibit the interaction between the protein and the transcription factor p53 [41], likewise results in the amelioration of clinical, neuropathological, and biomolecular parameters in animal models of relapsing remitting MS in EAE SJL mice, as indicated in [13]. Interestingly, PTM has been used to induce successful amelioration also in the cited animal models of neural disorders, where AA has also been shown to be active, supporting the possibility of converging mechanisms of action reasonably on S100B activity of the two drugs [42,43].

In any case, while the protective effect of AA may be reasonably considered to be related to S100B synthesis/secretion, a more general effect of this molecule on astrocytic physiopathology appears to be also suggested [19,33], although additional mechanisms at present cannot be ruled out. Astrocytes are known to play active roles in neurodegenerative disorders [39,44] and in MS in particular [45]; thus, a protective effect involving this cell type cannot be regarded as unexpected. Additional experiments will be needed to explore in detail possible effects of AA on this cell type, not only having in mind its effects on S100B synthesis/secretion, but also considering its heterogeneity [46].

## 4. Materials and Methods

### 4.1. Animal Procedure

Chronic EAE was induced in the C57Bl/6 strain of female mice (8–10-week-old) purchased from Charles River (USA). After anaesthesia, all mice were treated with immunogenic boosts composed by emulsion of myelin oligodendrocyte glycoprotein (MOG_35–55_, 50 mg/mL; Primm, Milan, Italy) and complete Freund’s adjuvant (CFA) used at 4× concentration containing 4 mg of heat-killed and dried Mycobacterium tuberculosis (strain H37Ra, ATTC 25177). Bordetella pertussis toxin (BDT) (Sigma-Aldrich S.r.l., Milan, Italy) was administered intraperitoneally (150 ng/mice) at day 0 and after 48 h. The procedure is described in Appendix A, according to the protocol of Stromnes and Goverman [47,48,49,50]. The mice were monitored daily for body weight and the development of clinical signs and symptoms (CSS; Appendix A) [51].

C57BL/6 mice were randomly distributed into four different groups: vehicle-treated healthy controls (CTRL + vehicle, *n* = 3), vehicle-treated EAE groups (EAE + vehicle, *n* = 16), AA-treated healthy controls (CTRL + AA, *n* = 3), and AA-treated EAE-affected mice (EAE + AA, *n* = 15). The mice were daily treated with the intraperitoneal injection of AA (4 mg/kg; ONO 2506, 50 mg; Tocris Bioscience, Milan Italy). The vehicle contained PBS and DMSO at same concentrations (0.5%) used to dissolve AA powder. The injections started at day 7. The mice were monitored daily for 24 days. Experiments were performed respecting ethical guidelines for animal welfare according to Università Cattolica del Sacro Cuore (Italian Ministry of Health, authorization *n* = 15/2021-PR protocol 1F295.120) whereby mice with premature disease severity and overt suffering were excluded and sacrificed before the end of the timepoint. The mice were sacrificed through perfusion with PBS, after deep intraperitoneal anaesthesia (87.5 mg/kg ketamine and 12.5 mg/kg xylazine, 0.1 mL/20 g body weight). CNS tissue was harvested, dissected into posterior (cerebellum, pons, bulb), anterior (prefrontal area), and cortex (rostral area) and stored at −80 °C for subsequent molecular analysis by q-PCR, ELISA, and quantization of ROS and NOS). Spinal cords were also harvested and immersed in a fixative solution of 4% paraformaldehyde (PFA) and then incubated for immunohistochemical and immunofluorescence analyses [52,53,54].

The animal procedures were performed in two different experiments. CNS derived from the first experiment were used for biomolecular and histologic analyses.

### 4.2. RT-qPCR Assay

Brain areas, collected as described above, were lysed through mechanical and enzymatic homogenization. Total RNA was extracted with an RNeasy mini kit following manufacturers’ instructions (Qiagen, Hilden, Germany). The cDNA obtained (sensiFast cDNA synthesis kit, Meridian Bioscience, Cincinnati, OH, USA) was used to quantify the expression level of *IL1β*, *INFγ*, and *S100B* in RT-qPCR, using an IQ SYBR^®^ Green supermix and iQ5 Multicolour Real Time PCR Detection System (Biorad, Hercules, CA, USA). β-actin was used as an internal control. The primer sequences used are detailed in Appendix A [13,55,56,57]. Conditions were described in our previous work [13]. Relative mRNA expression levels were calculated by normalizing on β-actin, using the 2^−^^Δ^^Ct^ method.

### 4.3. S100B ELISA Assay

S100B was quantified in the half-brain homogenates of EAE mice treated with AA, with the vehicle and the two respective control groups, by using a SimpleStep ELISA^®^ (enzyme-linked immunosorbent assay) kit (Abcam, Cambridge, United Kingdom). The kit was used according to the manufacturer’s instructions. S100B expression was evaluated by semi-quantitative analysis on C1–C4 cervical spinal cord segments, comparing EAE mice treated with the vehicle and AA. S100B levels were also measured in sera of wild type and EAE mice treated with the vehicle and AA. As expected, the sera S100B levels in the AA-treated EAE mice were lowered in comparison with those in the EAE mice treated with the vehicle, being comparable to levels in sera from wild-type mice (Appendix A)

### 4.4. Oxidative Stress Enzymes Activity

ROS were measured by an OxiSelect^TM^ Intracellular ROS Assay Kit (Green Fluorescence; Cell Biolabs, Inc. San Diego, CA, USA), according to the manufacturer’s instructions. NOS activity was measured by a colorimetric NOS Activity Assay Kit (BioVision, Kampenhout, Belgium), according to the manufacturer’s indication. All assays were performed on homogenates normalized for protein quantity (Biorad Assay Protein, Hercules, CA, USA).

### 4.5. Immunohistochemistry and Immunofluorescence

After fixation (4% PFA) and cryoprotection in 30% sucrose, spinal cords were cut with a cryostat (Leica CM1860 UV, Leica Biosystems, Milan, Italy). Serial sections (30 µm) from C1–C4 segments were stained for immunofluorescence and/or immunohistochemistry analysis to detect the following antigens: CD68 (Bio-Rad/AbD Serotec, CA, USA, 1:200), a marker of activated microglia, CD4+ (Bio-Rad/AbD Serotec, CA, USA, 1:50) for T-cell infiltrates, glial fibrillary acidic protein (GFAP; Cell Signalling Technology, MA, USA, 1:500), S100B (Novus biological, CO, USA, 1:1000) for astrocytes, and MBP (Cell Signalling Technology, MA, USA, 1:200) for myelin sheaths.

For immunohistochemistry after preincubation with 0.3% H_2_O_2_ in PBS, the sections were incubated at 4 °C with primary antibodies in a solution composed by PBS−0.3% Triton X-100−2% of normal donkey serum (NDS). Following the use of biotinylated donkey anti-mouse or donkey anti-rat antibodies (Jackson ImmunoResearch Europe Ltd., Ely, United Kingdom), avidin-biotin-peroxidase reactions were performed (Vectastain, ABC kit, Vector, Burlingame, CA, USA), using 3,3′-diaminobenzidine (Sigma-Aldrich, MI, Italy) as a chromogen. The sections were then analyzed using an Axioskop 2 optical microscope (Zeiss, Jena, Germany), with Neurolucida software (MBF Bioscience, Williston, VT, USA) for image acquisition.

For immunofluorescence, the sections were blocked with 10% NDS in 0.3% Triton X-100 in PBS, incubated with primary antisera/antibodies in 0.3% Triton X-100 and 2% NDS in PBS at 4 °C for 48 h and processed for immunofluorescence. The sections were carefully washed and incubated with proper fluorescent-conjugated secondary antibodies for 3 h at room temperature. The secondary antibodies in 0.3% Triton X-100 and 2% NDS in PBS were Alexa Fluor^®^ 488-AffiniPure donkey anti-mouse IgG (1:200, Jackson Immunoresearch; West Grove, PA, USA, in green), Alexa Fluor^®^ 488-AffiniPure donkey anti-rabbit IgG (1:200, Jackson Immunoresearch; West Grove, PA, USA, in green), Cy5-conjugated donkey anti-goat IgG (1:100, Jackson Immunoresearch; West Grove, PA, USA, in blue), and Cy3-conjugated donkey anti-rabbit IgG (1:100, Jackson Immunoresearch; West Grove, PA, USA, in red). After rinsing, the sections were mounted on slide glasses, covered with a fluoromount medium (Sigma-Aldrich, St Louis, MO, USA) and a coverslip and investigated with confocal microscopy. Immunofluorescence analysis was performed by a confocal laser scanning microscope (Zeiss, Jena, Germany, LSM 800) equipped with four laser lines, i.e., 405, 488, 561, and 639 nm. Brightness and contrast were adjusted with the Zen software 3.0 blue edition (Zeiss, Jena, Germany).

### 4.6. ImageJ Immunohistochemical Analysis

ImageJ software (version: 1.8.0_172, National Institutes of Health, Bethesda, MD, USA, https://imagej.nih.gov/ij/, accessed on 8 September 2021) analysis were performed for the evaluation of S100B, GFAP, CD4^+^, CD68^+^, and MBP distribution and presence into cervical C1–C4 spinal cord segments. Quantification was based on image acquisition through a ZEISS Microscope AXIO Imager 2 with a magnification of 5 (5×) and a magnification of 20 (20×). The acquired IHC images of S100B and GFAP cells and demyelination were evaluated trough IJMacro (https://imagej.nih.gov/ij/, accessed on 8 September 2021). This Software allowed us to investigate several plugins but select only markedly positive markers. Cell counts were based on several thresholds applied to the same image to allow the operator to choose the most suitable one (Image->Adjust->Autothreshold->operator choose). ImageJ software encoded the original image in a binary mask (0 for the background and 255 for positive cells). To decide the best threshold application, we compared all processed IHC images, and we selected the best ones. Indeed, for the S100B and GFAP cells positivity counts, we selected the MAXENTROPY threshold method, whereas for the MBP negativity areas, the OTSU one was chosen.

After the further exclusion of background noise (process->noise->”remove outliers”), a new ImageJ tool string (Watershed) allowed discerning among cell clusters. Moreover, considering that we were not interested in analyzing the grey matter of the spinal cord slices, we also inserted an ImageJ instrumental function in our algorithm program (freehand) to manually eliminate these uninterested areas. The counting tool (“Analyze Particles”, “size = 0-Infinity pixel show = Outlines summarize”) was run, only when binary images were appropriately focused for count.

The final content of the algorithm was written to create a way for the counter-proofing of results and to set a micrometer scale bar. We double-checked images through three computer commands (“multiply create”, “original”, and “processed”) and we set a suitable scale for the data output (“Set Scale”, “distance=; known=; unit = micron”). Our method validation confirmed that our counting algorithm did not differ consistently (percent standard deviation), comparing the script and manual methods (Appendix A).

In contrast to S100B^+^, GFAP^+^, and MBP, CD4^+^, and CD68^+^ cells were analyzed trough free-hand operator manual count (ImageJ tool). Algorithm applications were not available.

## Figures and Tables

**Figure 1 ijms-22-13558-f001:**
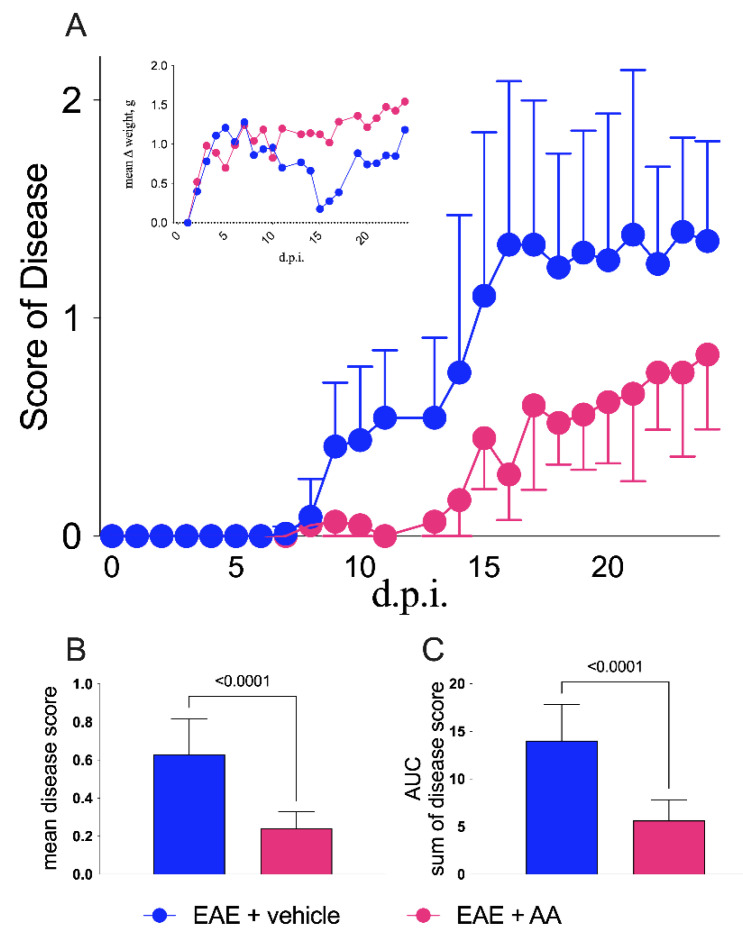
Clinical impact of arundic acid (AA) on experimental autoimmune encephalomyelitis (EAE). (**A**) Mice were monitored daily for clinical staging and sacrificed after 24 days post-immunization (d.p.i.). AA was used in the EAE-affected group (fuchsia line and symbols, *n* = 15) of mice that received a daily intraperitoneal injection of 4 mg/kg, i.e., 0.08 mL/day from 6 to 17 d.p.i., whereas in the control animal group, saline (PBS) was injected (blue line and symbols, *n* = 16). Beneficial effects of AA administration were significant from day 10 to day 24 (day 10: *p* = 0.04; day 11: *p* = 0.0003; day 13: *p* = 0.003; day 14: *p* < 0.0001; day 15: *p* < 0.0001; day 16: *p* < 0.0001; day 17 *p* < 0.0001; day 18: *p* < 0.0001; day 19: *p* < 0.0001; day 20: *p* < 0.0001; day 21: *p* < 0.0001; day 22: *p* = 0.009; day 23: *p* = 0.0002; day 24 0.007; two-way ANOVA corrected with the Bonferroni multiple comparison test). The left side insert shows the weight variations during the disease of the two groups of EAE mice, where the significant differences were detectable between 14 and 21 d.p.i. (day 14: *p* = 0.001; day 15: *p* = 0.0002; day 16: *p* < 0.0001; day 17: *p* = 0.007; day 19: *p* = 0.0002; day 20: *p* < 0.0001; day 21: *p* < 0.0001). Average disease score (**B**) and cumulative disease scores (**C**) summarized by bars comparing vehicle- (blue) and AA-treated EAE-affected animals; both graphs reveal significant differences in comparing the mean and the SEM between the two groups of mice (both with *p*-value < 0.0001, *t*-test, Mann−Whitney U test). Control groups (both the treated mice and the untreated healthy mice) are not displayed.

**Figure 2 ijms-22-13558-f002:**
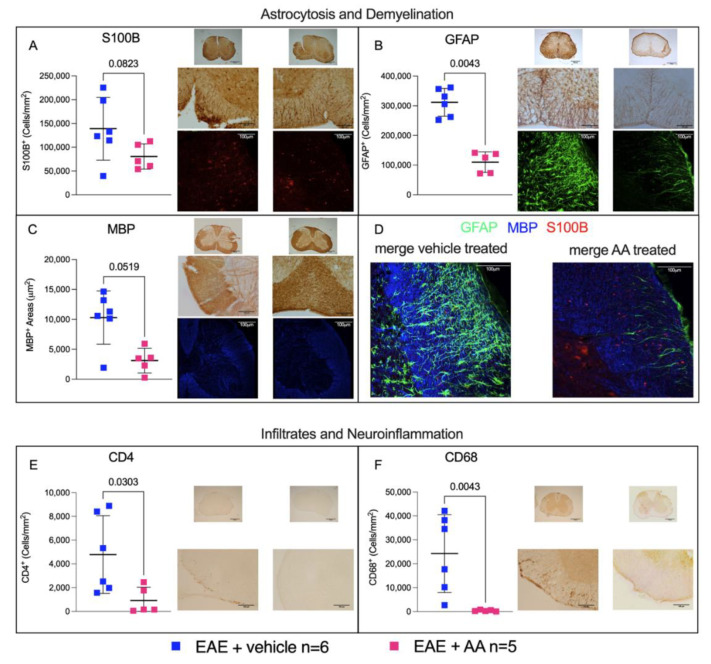
Histology of EAE-affected mice upon AA treatment. Immunochemistry (IHC) semi-quantification was obtained by ImageJ Software (National Institutes of Health, Bethesda, MD, USA, https://imagej.nih.gov/ij/) count for S100B^+^ and GFAP^+^ cells and MBP-negative areas, while CD4^+^ and CD68^+^ cells were manually counted. All counting results were obtained from C1–C4 spinal cord slices, comparing EAE-affected mice under different conditions of treatment: without (blue squares, *n* = 6, and left-sided graphs of each panel) or with AA (fuchsia squares, *n* = 5, and right-sided graphs of each panel). Images were reported following this order from the top to the bottom: IHC at 5× and 20× magnifications and Immunofluorescence (IF) at a 20× magnification (except for CD4 and CD68 staining where IF images are not available). Each image originated from a representative mouse of each group (vehicle- and AA-treated). (**A**–**D**) comparison of astrocytosis and demyelination through S100B (red signal), GFAP (green signal), and MBP (blue signal) staining and quantifications. (**B**) Comparison of GFAP^+^ cells/mm^2^ between the two groups of mice. The effect of AA on gliosis was analyzed with *t* test (Mann−Whitney, *p*-value = 0.043). The impact of AA on S100B was analyzed using *t* test (Mann−Whitney, *p*-value = 0.082). Demyelination rate is obtained from MBP negative areas sum (expressed in μm^2^ upon entire slice surface). The effect of AA demyelination is statistically consistent after *t* test (Mann−Whitney U test, *p*-value = 0.051, (**C**)). (**D**) Merging of the relative three immunofluorescence channel (S100B, MBP, and GFAP) of two representative mice (one from each group) at a 20× magnification. Of note, S100B immunostaining, not colocalizing neither with MBP nor with GFAP, was observed, possibly reflecting the S100B extracellularly secreted or the heterogeneous nature of astrocyte cell population, where S100B/GFAP colocalization was not constant [21]. (**E**,**F**) Immune infiltrates ((**E**); CD4^+^ cells/mm^2^) and neuroinflammation ((**F**); CD68^+^ cells/mm^2^). The impact of AA treatment was statistically significant (*p*-values = 0.0303 and 0.0043 for *t* test and Mann−Whitney U test, respectively). The representative images related to each group of treatment were reported in 5× and 20× magnifications (IHC).

**Figure 3 ijms-22-13558-f003:**
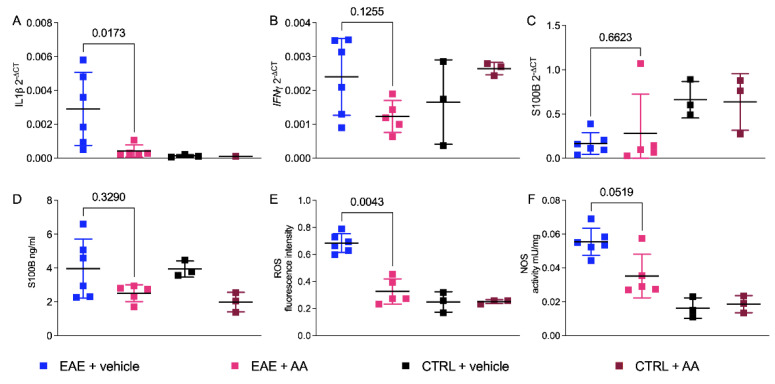
Effects of AA on inflammatory cytokines, S100B, and oxidative stress enzymes. (**A**–**C**) Comparison of gene expression levels (2^−∆CT^) of inflammatory cytokines and S100B among different groups of mice: treated (AA; fuchsia squares) and untreated (vehicle; blue squares) EAE, and treated (AA; magenta squares) and untreated (vehicle; black squares) controls. From (**A**), it can be seen that the reduction of IL1β in EAE-affected mice treated with AA was statistically significant (*p* = 0.0173, *t*-test, Mann−Whitney U test). IFNγ (**B**) and S100B (**C**) expression levels which did not present differences between the AA- and vehicle-treated EAE groups. (**D**) S100B protein levels (nanograms/milliliters) resulting in a trend of reduction in the AA-treated EAE group, although not statistically significant. ROS ((**E**); *p*-value = 0.0043, *t*-test, Mann−Whitney U test) and NOS ((**F**); *p*-value = 0,0519, *t*-test, Mann−Whitney U test) activities displayed as fluorescence intensity and as milliunits/milligrams, respectively. In (**E**,**F**), oxidative stress enzymes appeared to be significantly reduced by AA treatment during EAE. The statistical analyses comparing the EAE-affected and control groups are not displayed.

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
