# Peer review of "S100B Protein as a Therapeutic Target in Multiple Sclerosis: The S100B Inhibitor Arundic Acid Protects from Chronic Experimental Autoimmune Encephalomyelitis"

_ijms, 2021, doi:10.3390/ijms222413558_

Round 1

Reviewer 1 Report

  • Please describe clinical disease scoring in detail in supplementary as the results in Fig 1 highlight significant differences in clinical score in the AA treated group. What parameters were involved in the clinical scoring? How many independent assessors assessed this data set?

  • Please label either the y axis of all the graphs in Fig 2 to help the readers as to what types of comparison are presented example: label like this GFAP+ cells/mm2

Author Response

Reviewer 1

  • Please describe clinical disease scoring in detail in supplementary as the results in Fig 1 highlight significant differences in clinical score in the AA treated group. What parameters were involved in the clinical scoring? How many independent assessors assessed this data set?

As requested by both Reviewers we detailed the assessment of clinical disease activity Supplementary Table 2 that summarizes the scores. Briefly: the induction of EAE was performed by an operator (GDS) who divided mice into treated and control groups, assigning different codes, dividing them into different cages and distinguishing animals with ear holes. This operator was the only one in charge of the daily treatments, of the daily registration of the scores and of the supervision of both experiments. Clinical disease score was assessed by 3 operators (CC, MDC and MT), who performed alone the double-blind examinations of the animals, ignoring both codes and treatments and the score of the other operators. The clinical assessment was performed at the same hours twice everyday (8.30-9.00 a.m. and 3.00-3.30 p.m.) except for Saturday and Sunday where it was performed only in the morning. The operators rotated each day so that the one who had visited one day in the morning would have visit in the afternoon shift of the next day and viceversa, in order to avoid any bias depending on the circadian rhythm of the animal and on the routine of the visits. All the operators were trained by GDS in previous experiments and had at least 1 year experience with EAE affected mice handling. As described in the table inserted in the Supplementary Table 2, a 0 to 3.5 score was used, according to the animal welfare of Health Ministry of Italy that impose a termination criterium when clinical disease score is more severe than score 3. In detail the scores are as following:  0 = healthy; 0.5 = weak tail or unsteady gait; 1 = Limp tail or weak tail and hindleg weakness/unsteady gait; 1.5 = Unilateral hindlimb weakness; 2 = Bilateral hindlimb weakness; 2.5 = Unilateral hindlimb paraplegia; 3 = Bilateral hindlimb paraplegia; 3.5 = Moderate loss of lower body control. The score resulted from the mean calculated by the operator in charge of registration of data (GDS) on the basis of the scores assessed by the different visiting operators for each day, except for the weekend, when the evaluation was performed just in the morning.

  • Please label either the y axis of all the graphs in Fig 2 to help the readers as to what types of comparison are presented example: label like this GFAP+cells/mm2

As suggested, we emendated the Figure 2.

Reviewer 2 Report

The manuscript ‘S100B protein as a therapeutic target in multiple sclerosis: the S100B inhibitor arundic acid protects from chronic experimental autoimmune encephalomyelitis’ aims to study the effect of arundic acid which is an inhibitor of S100b protein and find if the treatment with arundic acid can ameliorate the symptoms of neuropathology in the animal model of Multiple Sclerosis.

Although the author presented good work, there are some concerns that need to be clarified.

Major concern:

  1. Can the author describe the scoring system for EAE? Although significant but there is a very small difference between the two treatment groups. What stringent scoring system was used to evaluate these differences?
  2. Fig.2 is the result of the histology analysis of treated EAE mice. Although the graphical results are clear there are a few concerns about the representative images.
  • The figures need to be properly labeled since it is not possible to detect differences between the two treatment groups.
  • A and C: Due to differences in the background of 5X IHC images, it is not possible to justify the difference. Can the author provide images with a similar background?
  • Fig 2E and F: The difference between treatment groups can not be appreciated for both 5X and 20X images since it is blue in color. Can the author provide better/clear images without any contrast?
  • Please provide a scale bar for each image in fig.2
  1. In fig.2D, AA-treated mice show distinct expressions of S100B which do not localize with either MBP (oligodendrocytes) or GFAP (astrocytes). Can the author discuss why a majority of secreted S100B accumulated in that region of the spinal cord?

4. The author has examined the cervical spinal cord by IHC (fig.2) and quantified inflammation (fig.3) in the posterior regions of the brain

  • Can the author explain the reason why they have selected different regions of CNS to study EAE?
  • Can the author explain the term, ‘posterior region of CNS’ and tell us the exact location of CNS used to study fig 3?
  • Can we expect differences in inflammation in different regions of CNS and how comparing two regions of CNS can be justified? Do you expect a difference in expression of S100 in the cervical spinal cord or any other regions of the CNS? 

Minor comments:

  1. Line 66- Explain the word ‘EF’.
  2. Line 147- CD68+ cells can be both reactive microglia and circulating monocytes recruited to the CNS during MS

Author Response

Reviewer 2

The manuscript ‘S100B protein as a therapeutic target in multiple sclerosis: the S100B inhibitor arundic acid protects from chronic experimental autoimmune encephalomyelitis’ aims to study the effect of arundic acid which is an inhibitor of S100b protein and find if the treatment with arundic acid can ameliorate the symptoms of neuropathology in the animal model of Multiple Sclerosis. Although the author presented good work, there are some concerns that need to be clarified. Major concern:

  1. Can the author describe the scoring system for EAE? Although significant but there is a very small difference between the two treatment groups. What stringent scoring system was used to evaluate these differences?

As requested by both Reviewers we detailed the assessment of clinical disease activity adding to Supplementary Table 2 a table that summarize the scores. Briefly: the induction of EAE was performed by an operator (GDS) who divided mice into treated and control groups, assigning different codes, dividing them into different cages and distinguishing animals with ear holes. This operator was the only one in charge of the daily treatments, of the daily registration of the scores and of the supervision of both experiments. Clinical disease score was assessed by 3 operators (CC, MDC and MT), who performed alone the double-blind examinations of the animals, ignoring both codes and treatments and the score of the other operators. The clinical assessment was performed at the same hours twice everyday (8.30-9.00 a.m. and 3.00-3.30 p.m.) except for Saturday and Sunday where it was performed only in the morning. The operators rotated each day so that the one who had visited one day in the morning would have visit in the afternoon shift of the next day and viceversa, in order to avoid any bias depending on the circadian rhythm of the animal and on the routine of the visits. All the operators were trained by GDS in previous experiments and had at least 1 year experience with EAE affected mice handling. As described in the table inserted in the Supplementary Table 2, a 0 to 3.5 score was used, according to the animal welfare of Health Ministry of Italy that impose a termination criterium when clinical disease score is more severe than score 3. In detail the scores are as following:  0 = healthy; 0.5 = weak tail or unsteady gait; 1 = Limp tail or weak tail and hindleg weakness/unsteady gait; 1.5 = Unilateral hindlimb weakness; 2 = Bilateral hindlimb weakness; 2.5 = Unilateral hindlimb paraplegia; 3 = Bilateral hindlimb paraplegia; 3.5 = Moderate loss of lower body control. The score resulted from the mean calculated by the operator in charge of registration of data (GDS) on the basis of the scores assessed by the different visiting operators for each day, except for the weekend, when the evaluation was performed just in the morning.

  1. Fig.2 is the result of the histology analysis of treated EAE mice. Although the graphical results are clear there are a few concerns about the representative images.

  • The figures need to be properly labeled since it is not possible to detect differences between the two treatment groups.

As suggested, we emendated the Figure 2.

  • A and C: Due to differences in the background of 5X IHC images, it is not possible to justify the difference. Can the author provide images with a similar background?

As suggested, we reacquired new images by regulating the light of exposure and the contrast in order to standardize the background.

  • Fig 2E and F: The difference between treatment groups can not be appreciated for both 5X and 20X images since it is blue in color. Can the author provide better/clear images without any contrast?

As for the previous response we reacquired the images avoiding, as suggested, any contrast.

  • Please provide a scale bar for each image in fig.2

As suggested we emendated Figure 2 adding scale bars where they were missing.

  1. In fig.2D, AA-treated mice show distinct expressions of S100B which do not localize with either MBP (oligodendrocytes) or GFAP (astrocytes). Can the author discuss why a majority of secreted S100B accumulated in that region of the spinal cord?

We thank the Reviewer for this interesting consideration. It is in fact reasonable that a significant aliquot of S100B is secreted during EAE processes and, as a consequence, may be detected using immunohistochemical procedures in neural tissue - in this specific case in the spinal cord - when it is the site of an active inflammatory lesion, reasonably independently from regional peculiarities.

In addition, but not as a secondary consideration, the observation of S100B not colocalizing neither with MBP nor with GFAP may be attributed to the recognized heterogeneous nature of the astrocyte cell population. In fact, S100B and GFAP are regarded not to colocalize in all astrocytes, being observed subpopulations of these cells containing only S100B or only GFAP (Steiner et al, 2007). This information has been added in the Legend to Fig 2 (Page 6, Lines 223-226, clean version of the revised manuscript)

  1. Steiner, J.; Bernstein, H.-G.; Bielau, H.; Berndt, A.; Brisch, R.; Mawrin, C.; Keilhoff, G.; Bogerts, B. Evidence for a Wide Extra-Astrocytic Distribution of S100B in Human Brain. BMC Neurosci 2007, 8, 2, doi:10.1186/1471-2202-8-2.

4. The author has examined the cervical spinal cord by IHC (fig.2) and quantified inflammation (fig.3) in the posterior regions of the brain

  • Can the author explain the reason why they have selected different regions of CNS to study EAE?

We thank the Reviewer for giving us the opportunity of better clarifying our choices regarding mice neural tissues used for different experimental sets.

We examined spinal cord for immunohistochemical studies since in this neural compartment neuropathological markers are known to be especially apparent in mice EAE (for review, Burrows et al, 2019). This consideration has been added in Results section (Page 3, Lines 133-134 and Line 141, clean version of the revised manuscript). We showed biomolecular parameters in a different neural district (“posterior” brain region) in order to offer a wide information concerning EAE nervous system in AA treated and untreated mice. However, we also examined, using immunohistochemistry, also another area of the CNS (cerebellum), where these differences were partially detectable, although less evident, as expected. These results are shown in the revised version of this manuscript as Supplementary Figure 3. Likewise, we also tested IL1b and S100B gene expression levels from mRNA derived from spinal cords (C1-C4 segments) of vehicle- vs AA-treated EAE affected mice, finding results comparable to cerebellum molecular data and immunohistochemical spinal cord outcomes. We added these data to the Supplementary Figure 4 (Page 4, Lines 178-181, clean version of the revised manuscript).

  • Can the author explain the term, ‘posterior region of CNS’ and tell us the exact location of CNS used to study fig 3?

In this manuscript the term “posterior region of CNS” indicates cerebellum, pons and bulb, as indicated defined in the Materials and Methods section (Page 9, line 339, clean version of the revised manuscript). Thanks to the Reviewer’s comment we better clarified this in the text (Page 4, Lines 166-167, clean version of the revised manuscript).

  • Can we expect differences in inflammation in different regions of CNS and how comparing two regions of CNS can be justified?

As indicated in a previous item, both for immunohistochemical and biomolecular analyses corresponding neural districts (spinal cord or “posterior” brain) were examined in treated and untreated mice, so that different CNS regions were never in fact compared. However, in the present revised version we also showed immunohistochemistry in the cerebellar region (Supplementary Fig 3) and biomolecular results concerning S100B and IL1b in C1-C4 segments (Supplementary fig 4) (Page 4, Lines 178-181, clean version of the revised manuscript).

  • Do you expect a difference in expression of S100 in the cervical spinal cord or any other regions of the CNS?

The distribution of S100B in different neural districts of wild type vertebrates has been widely investigated since the discovery of the protein, sometimes with conflicting results. Recently, in C57Bl/6 mice a higher expression of S100B in the “posterior” regions of the brain has been described (Hagmeyer et al, 2019), especially in the cerebellum, which is the neural district tested in this work for biomolecular assays, including S100B evaluation. We added this consideration in the revised version of the manuscript (Page 4, Line 166, clean version of the revised manuscript). However, the S100B distribution in wild type mice, although interesting in general terms for the possible biological role of the protein, might be limitedly relevant in this study (Page 8, Lines 277-278, clean version of the revised manuscript).

  1. Hagmeyer, S.; Romão, M.A.; Cristóvão, J.S.; Vilella, A.; Zoli, M.; Gomes, C.M.; Grabrucker, A.M. Distribution and Relative Abundance of S100 Proteins in the Brain of the APP23 Alzheimer’s Disease Model Mice. Front. Neurosci. 2019, 13, 640, doi:10.3389/fnins.2019.00640.

Minor comments:

  1. Line 66- Explain the word ‘EF’.

EF stands for helix-loop-helix structure that is formed by the E and F helices (letters assigned to helices in the order that they occur, starting at the N-terminus). We included this definition (Page 2, Lines 66-67).

  1. Line 147- CD68+ cells can be both reactive microglia and circulating monocytes recruited to the CNS during MS

We emendated as suggested (Page 4, Lines 151-2, clean version of the revised manuscript).

Round 2

Reviewer 2 Report

Appreciate the response to the comments and for making the necessary changes.